# Potential Predictability of Seasonal Global Precipitation Associated with ENSO and MJO

Haibo Liu [1], Xiaogu Zheng [2,*], Jing Yuan [3] and Carsten S. Frederiksen [4,5,6]

[1] Lamont-Doherty Earth Observatory, Columbia University, Palisades, NY 10964, USA; haibo@ldeo.columbia.edu
[2] International Global Change Institute, Hamilton 3210, New Zealand
[3] International Research Institute for Climate and Society (IRI), Columbia University, Palisades, NY 10964, USA
[4] The Bureau of Meteorology, Melbourne, VIC 3001, Australia
[5] The School of Earth, Atmosphere and Environment, Monash University, Clayton, VIC 3800, Australia
[6] CSIRO, Oceans and Atmosphere, Aspendale, Melbourne, VIC 3195, Australia
[*] Correspondence: xiaogu.zheng@gmail.com or xiaogu.zheng@igci.org.nz

**Abstract:** A covariance decomposition method is applied to a monthly global precipitation dataset to decompose the interannual variability in the seasonal mean time series into an unpredictable component related to "weather noise" and to a potentially predictable component related to slowly varying boundary forcing and low-frequency internal dynamics. The "potential predictability" is then defined as the fraction of the total interannual variance accounted for by the latter component. In tropical oceans (30° E–0° W, 30° S–30° N), the consensus is that the El Nino-Southern Oscillation (ENSO, with 4–8 year cycles) is a dominant driver of the potentially predictable component, while the Madden-Julian Oscillation (MJO, with 30–90 days cycles) is a dominant driver of the unpredictable component. In this study, the consensus is verified by using the Nino3-4 SST index and a popular MJO index. It is confirmed that Nino3-4 SST does indeed explain a significant part of the potential predictable component, but only limited variability of the unpredictable component is explained by the MJO index. This raises the question of whether the MJO is dominant in the variability of the unpredictable component of the precipitation, or the current MJO indexes do not represent MJO variability well.

**Keywords:** predictability; global; seasonal precipitation; ENSO; MJO

## 1. Introduction

The seasonal mean time series of meteorological variables are widely used for analyzing interannual climate variability and predictability. Studies of potential predictability are usually based upon a decomposition of the seasonal variability into a part called the "weather noise" variability, which is fundamentally unpredictable beyond a deterministic time period, and another part assumed to be at least potentially predictable [1]. The potential predictability is measured as the fraction of the total variability accounted for by the latter part. It helps us to identify regions where the potential for making skillful climate forecasts is highest, as well as where climate noise dominates any signal. Such knowledge is useful to investigate the roles of drivers of potentially predictable variability, such as human-induced forcing, oceanic and slowly varying atmospheric states, soil moisture and snow cover conditions, and the roles of stochastic meteorological processes generated by typical weather events in modifying regional variability. They can also help provide guidance as to where and when the issuance of climate forecasts may be most reasonable.

The measure of potential predictability is clearly sensitive to how the separation of the variance is performed. Some may think that temporal filtering techniques could be employed based on the assumption that weather noise operates mainly on timescales much shorter than that of the potentially predictable variability [2]. However, weather

events include not only high-frequency day-to-day fluctuations, but also intra-seasonal fluctuations (such as MJO) which cannot be completely smoothed out by a seasonal mean filter. The residuals through the seasonal mean filter give rise to chaotic, unpredictable fluctuations in the seasonal mean time series. Therefore, it is not possible to completely isolate the potentially predictable variability through temporal filtering.

In this paper, the methodology proposed by Zheng and Frederiksen [3] is applied to analyze the potential predictability of a monthly global precipitation data. It was found that for rainfall in tropical oceans, the main driver of predictable variability is ENSO, while the unpredictable variability is related to MJO, but not as dominant as ENSO to the predictable variability. Regions with significant potential predictability are also identified. The methodology applied in this paper and other methodologies are reviewed.

The current paper begins with a description of the data and the estimation method of the potential predictability in Section 2. Section 3 is devoted to applications to monthly global precipitation data. The methodology applied in this paper and other methodologies are reviewed in Section 4. A summary is presented in Section 5.

## 2. Data and Methods

### 2.1. Data

The data used in this study include monthly precipitation data (on a $2.5° \times 2.5°$ grid) from the Global Precipitation Climatology Project (GPCP), version 2.3 [4], for the period $1979-2020$, which is a widely used global (land and ocean) precipitation dataset derived from a mix of satellite estimates over oceans and land, and rain gauge measurements from land and atolls. The GPCP is often used to study variations in precipitation at global and regional scales (https://climatedataguide.ucar.edu/climate-data/gpcp-monthly-global-precipitation-climatology-project#:~:text=The%20GPCP%20monthly%20dataset%20is,with%20some%20delay%20for%20processing) (accessed on 6 April 2023). The Nino3-4 index is calculated from the HadISST1 SST as the average of SST anomalies over the region $5° N$–$5° S$ and $170°$–$120° W$. The MJO index used here is the outgoing longwave radiation-based MJO index (OMI) (https://www.psl.noaa.gov/mjo/mjoindex/) (accessed on 6 April 2023). The OMI calculation is based on an empirical orthogonal function (EOF) analysis. The two components of OMI, MJO-PC1 and MJO-PC2, are the projections of 20–96 days filtered outgoing longwave radiation onto the first and the second daily spatial EOF patterns of 30–96 days eastward filtered OLR. The OMI can successfully capture the convective component of the MJO [5].

### 2.2. Identifying the Seasonal Predictable and Unpredictable Components

The (co-)variance decomposition method of Zheng and Frederiksen [3] is proposed for deriving the spatial patterns of interannual variability in the seasonal mean fields related to the variability of predictable and unpredictable components, based on monthly mean data. First, the annual cycle is removed from the data by subtracting the multi-year mean for each month. This serves to reduce the climate bias in the covariance estimation. Then, a conceptual statistical model for a climate variable $X_{ym}$ in month $m$ ($m$ = 1, 2, 3 in a specific season, e.g., June, July, August) and in year $y$ ($y$ = 1, . . . , $Y$, where $Y$ is the total number of years) expressed as per the following regression form:

$$X_{ym} = \mu_y + \varepsilon_{ym} \tag{1}$$

where $\mu_y$ (interception) represents the seasonal "statistical population" mean in year $y$; and $\varepsilon_{ym}$ is the residual monthly departure of $X_{ym}$ from $\mu_y$. Then, the seasonal mean of a climate variable in year $y$ ($X_{yo} \equiv \frac{1}{3}\sum_{m=1}^{3} X_{ym}$) can be conceptually expressed as

$$X_{yo} = \mu_y + \varepsilon_{yo} \tag{2}$$

where $\varepsilon_{yo} \equiv \frac{1}{3}\sum_{m=1}^{3} \varepsilon_{ym}$ is the seasonal mean of the monthly (also daily) residuals for $\mu_y$ that arises from intra-seasonal variability ($\varepsilon_{ym}$) and is not smoothed out by the seasonal

mean filter. Since the weather system is chaotic, $\varepsilon_{ym}$ is unlikely to be predictable beyond one or two weeks, so $\varepsilon_{yo}$ is essentially unpredictable beyond the deterministic prediction period and is referred to as the unpredictable component of $X_{yo}$. On the other hand, the interception $\mu_y$ is a constant through the season in $y$ year and is more likely dominated by the external forcing and slowly varying internal dynamics. Therefore, $\mu_y$ is potentially predictable beyond the deterministic prediction period and is referred to as the potentially predictable component of $X_{yo}$ [1]. It is very difficult to numerically separate $\mu_y$ from the seasonal mean $X_{yo}$ using filter techniques, because $X_{yo}$ is already the best estimation of $\mu_y$, but subject to error $\varepsilon_{yo}$.

Let $\left\{ X'_{ym}, y = 1, \ldots, Y; m = 1, 2, 3 \right\}$ be another climate variable in the form as Equation (1), then, using the statistical technique based on the (co-)variance decomposition [4], the interannual covariance of the unpredictable components can be calculated with monthly data by using Equation (16) in Zheng and Frederiksen [3] as follows:

$$V\left(\varepsilon_{yo}, \varepsilon'_{yo}\right) = \hat{\sigma}^2(3 + 4\hat{\varphi})/9 \tag{3}$$

where

$$\hat{\sigma}^2 = \frac{a}{2(1 - \hat{\varphi})} \tag{4}$$

$$\hat{\varphi} = \min\left\{0.1, \max[(a + 2b)/(2a + 2b), 0]\right\} \tag{5}$$

$$a = \frac{1}{2}\left\{\frac{1}{Y}\sum_{y=1}^{Y}\left[X_{y1} - X_{y2}\right]\left[X'_{y1} - X'_{y2}\right] + \frac{1}{Y}\sum_{y=1}^{Y}\left[X_{y2} - X_{y3}\right]\left[X'_{y2} - X'_{y3}\right]\right\} \tag{6}$$

and

$$b = \frac{1}{2}\left\{\frac{1}{Y}\sum_{y=1}^{Y}\left[X_{y1} - X_{y2}\right]\left[X'_{y2} - X'_{y3}\right] + \frac{1}{Y}\sum_{y=1}^{Y}\left[X_{y2} - X_{y3}\right]\left[X'_{y1} - X'_{y2}\right]\right\} \tag{7}$$

The interannual covariance of the predictable components $V\left(\mu_y, \mu'_y\right)$ can be calculated as

$$V\left(\mu_y, \mu'_y\right) = V\left(X_{yo}, X'_{yo}\right) - V\left(\varepsilon_{yo}, \varepsilon'_{yo}\right) \tag{8}$$

where $V\left(X_{yo}, X'_{yo}\right)$ represents the total interannual covariance and can be estimated directly from seasonal mean variables $X_{yo}$ and $X'_{yo}$, i.e.,

$$V\left(X_{yo}, X'_{yo}\right) = \frac{1}{Y}\sum_{y=1}^{Y} X_{yo} X'_{yo} \tag{9}$$

The R-code for calculating the unpredictable covariance of Equation (3) is included in Appendix A. The $\chi^2$ test and student's $t$-test, used to judge the significance of the interannual covariances of the predictable and unpredictable components, i.e., Equations (3) and (8) respectively, are documented in the Appendix of Ying et al. ([6]; their Equations (10)–(14)).

The potential predictability is defined as the ratio of the predictable variance against the total variance, i.e.,

$$V\left(\mu_y, \mu_y\right)/V\left(X_{yo}, X_{yo}\right) \tag{10}$$

The significant test procedure for the potential predictability is documented in Equation (8) of Zheng et al. [7].

Define the standardised predictable and unpredictable covariances for the $X'_{yo}$ with $X_{yo}$ as

$$V\left(\mu'_y, \mu_y\right)/\sqrt{V\left(\mu'_y, \mu'_y\right)} \tag{11}$$

and

$$V\left(\varepsilon'_{yo}, \varepsilon_{yo}\right) \Big/ \sqrt{V\left(\varepsilon'_{yo}, \varepsilon'_{yo}\right)} \tag{12}$$

respectively. It can be proved that the square of Equation (11) (Equation (12)) is the variance of the predictable (unpredictable) component of $X_{yo}$ explained by the predictable (unpredictable) component of $X'_{yo}$ (see Appendix B for proof). Therefore, the fraction of the predictable (unpredictable) component of $X_{yo}$ explained by the predictable (unpredictable) component of $X'_{yo}$ is the square of predictable (unpredictable) correlation.

$$\frac{V^2\left(\mu_y, \mu'_y\right)}{V(\mu_y, \mu_y)\,V\left(\mu'_y, \mu'_y\right)} = \left[\frac{V\left(\mu_y, \mu'_y\right)}{\sqrt{V(\mu_y, \mu_y)\,V\left(\mu'_y, \mu'_y\right)}}\right]^2 \triangleq cor^2(\mu_y, \mu'_y) \tag{13}$$

and

$$\frac{V^2\left(\varepsilon_{yo}, \varepsilon'_{yo}\right)}{V(\varepsilon_{yo}, \varepsilon_{yo})\,V\left(\varepsilon'_{yo}, \varepsilon'_{yo}\right)} = \left[\frac{V\left(\varepsilon_{yo}, \varepsilon'_{yo}\right)}{\sqrt{V(\varepsilon_{yo}, \varepsilon_{yo})\,V\left(\varepsilon'_{yo}, \varepsilon'_{yo}\right)}}\right]^2 \triangleq cor^2(\varepsilon_{yo}, \varepsilon'_{yo}) \tag{14}$$

Moreover, if another climate variable $X''_{yo}$ is statistically independent of $X'_{yo}$, then the fraction of the predictable (unpredictable) component of $X_{yo}$ explained by the predictable (unpredictable) components of $X'_{yo}$ and $X''_{yo}$ is the sum of those for $X'_{yo}$ and $X''_{yo}$, respectively.

## 3. Results

In principle, predictable variability is driven by the slowly varying external forcing and slowly varying internal dynamics which are stable within a season, while unpredictable variability is driven by the internal dynamics dominated by intra-seasonal variability with cycles larger than 10 days. In this section, the impacts of ENSO and MJO on the predictable and unpredictable variabilities of global precipitation, especially on tropical oceans, are studied.

### 3.1. El Nino-Southern Oscillation

In this study, the Nino3-4 index is used to represent the ENSO. The potential predictability of the index is more than 0.9 for all seasons indicating that the predictable variability is dominant. Therefore, any global precipitation related to ENSO is also likely to be highly predictable.

Figure 1 shows that the variability in the predictable component of precipitation is largest in the tropics in all seasons, especially in DJF. There is a clear seasonal cycle, with the largest predictable variability in DJF and the smallest variability in JJA. This coincides with the seasonal cycle of ENSO, which is also the strongest in DJF and the weakest in JJA. In the peak season DJF, the center of the largest predictable variability is in the Nino3-4 region (5° N–5° S, 170° W–120° W). This may be due to the fact that in the La Nina phase, the cold equatorial water upwells in the Nino3-4 region, associated with a strong trade wind passing through the Nino3-4 region, resulting in it being the driest; while in the El Nino phase, the sea surface temperature in the Nino3-4 region is the warmest, resulting in a large amount of convective cloud that leads to significant positive rainfall anomalies in the Nino3-4 region. All these facts indicate that ENSO is a likely dominant driver of the predictable variability of tropical precipitation. To further verify this point, the standardized predictable covariances for the Nino3-4 index with global precipitation, in all four seasons, are shown in Figure 2. Regardless of their signs, the spatial patterns and strength of their absolute values are very similar to those of the predictable component variability (Figure 1) in tropical oceans (30° E–0° W, 30° S–30° N). This is confirmed by the fact that the pattern correlations between the predictable component standard deviations (Figure 1) and the absolute values of the standardized predictable covariances for GPCP precipitation with

Nino3-4 SST (Figure 2) in the tropical oceans are 0.79, 0.82, 0.89 and 0.92 in MAM, JJA, SON and DJF, respectively. The corresponding means of the fraction of the predictable precipitation variances explained by the predictable component of Nino3-4 SST (the square of Equation (10)) over the tropical oceans are 0.24, 0.41, 0.59, 0.56, respectively. In SON and DJF when ENSO is stronger and unlikely to change phase, the correlations and explained predictable variances are more significant than those in MAM and JJA. Although the ENSO variability in MAM is stronger than that in JJA, the correlation and the explained predictable variance are less significant than those in JJA. This coincides with the fact that ENSO is more likely to change phase in MAM (especially in April) than in JJA.

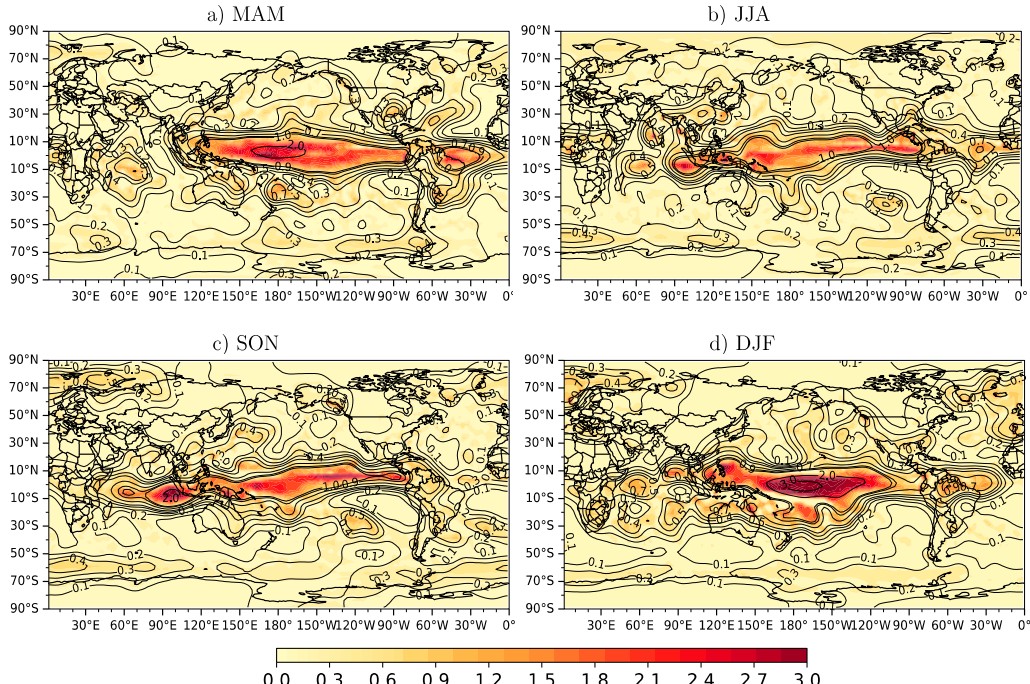

**Figure 1.** Predictable component standard deviation of GPCP precipitation (mm/day).

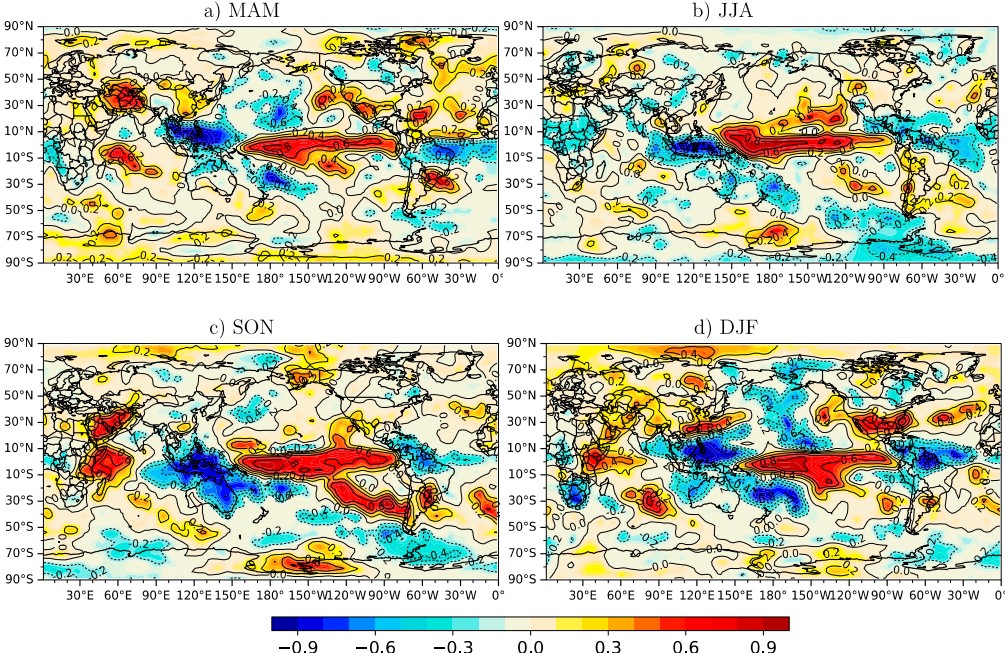

**Figure 2.** Standardized predictable covariances for GPCP precipitation with Nino3-4 SST.

### 3.2. Madden-Julian Oscillation

The Madden-Julian Oscillation (MJO) is a tropical weather system originating in the Indian Ocean that propagates eastward around the global tropics with a cycle on the order of 30–90 days [8]. The MJO indices MJO-PC1 and MJO-PC2 are the projections of 20–96 days filtered OLR [5], the seasonal potential predictability of the MJO-PC1 and MJO-PC2 are virtual zero. Thus, the MJO is more likely to be a driver of the unpredictable variability of tropical precipitation. Therefore, any global precipitation related to the two MJO indices is also likely to be highly unpredictable on a seasonal time scale.

The MJO consists of two parts; one that has strong convective rainfall (wet) and the other with suppressed rainfall (dry). The dry part of the MJO always precedes the wet part (dry in the east, wet in the west). There are less clouds in the dry part, which induces strong solar heating reaching the ocean surface underneath. The accumulation of the heating on the ocean surface causes warmer SST, which in turn causes upward movement of the air. This results in convection (wet part) moving eastward, and hence the east propagation of the MJO. The MJO has wide ranging impacts on the patterns of tropical and extratropical precipitation, atmospheric circulation, and surface temperature around the global tropics and subtropics.

We compare the seasonal cycles of the MJO described by Zhang and Dong [8] with the unpredictable variability of precipitation (Figure 3) and the standardized unpredictable covariance for the MJO index with global precipitation (Figure 4) in tropical oceans. Zhang and Dong [8] confirmed that the primary peak season is in boreal winter (December–March), during which MJO signals are mainly confined to the Indian Ocean and western Pacific Ocean, and reach their maxima in the South Pacific convergence zone. Similarly, Figure 2d shows that the primary peak season of unpredictable variability is in DJF with the most significant variability near the equatorial Indian Ocean (with standard deviation 1.5 mm/day) and western Pacific Oceans, especially in the South Pacific convergence zone (with standard deviation 2.0 mm/day). Figure 3a shows the unpredictable variability in MAM with a similar pattern to DJF, but with less variability. This coincides with the fact that March also belongs to the MJO peak season. This is also the case for the standardized unpredictable covariances for GPCP precipitation with MJO index (Figure 4d) in the tropical ocean, but with much weaker strength (with standard deviation 0.8–1.0 mm/day in the Indian Ocean and 0.6–1.0 mm/day in the South Pacific convergence zone).

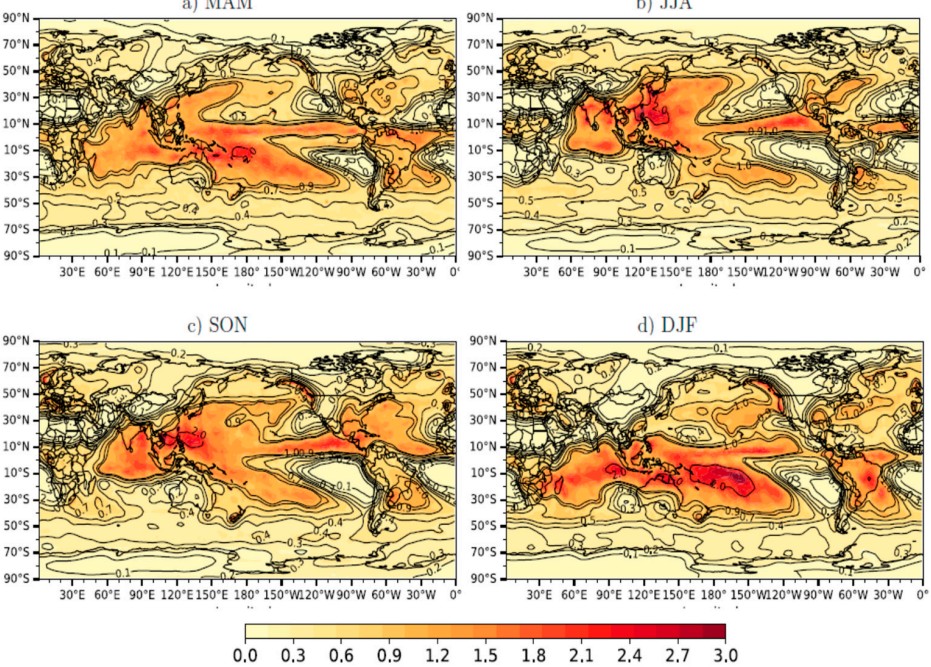

**Figure 3.** Unpredictable component standard deviation of GPCP precipitation (mm/day).



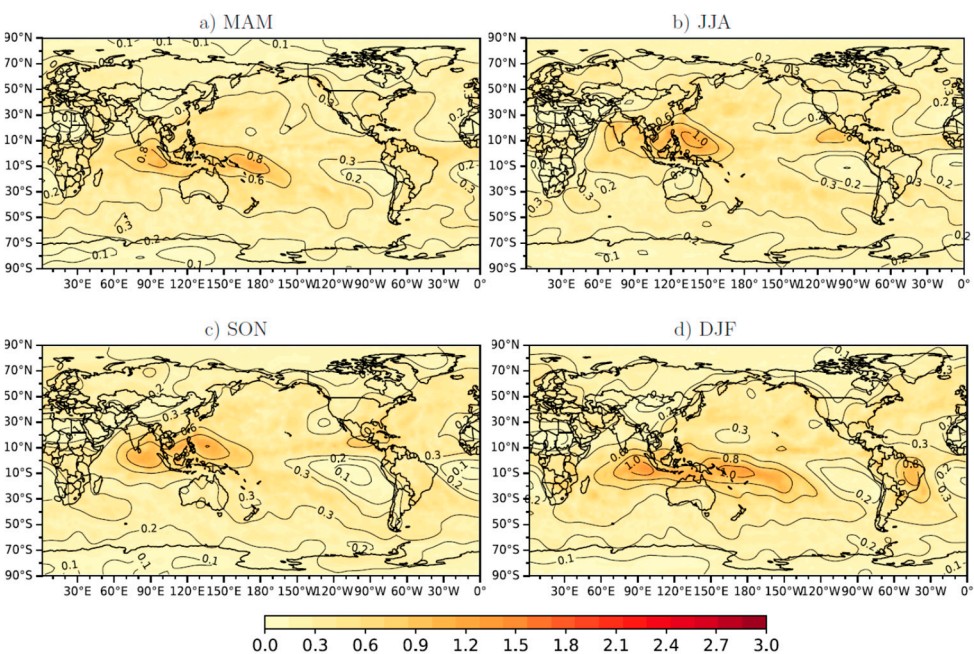

**Figure 4.** Standardized unpredictable covariances for GPCP precipitation with MJO index.

Zhang and Dong [8] further confirmed that the secondary peak season of the MJO is boreal summer (June–September), during which the strongest MJO occurs north of the equator from the Bay of Bengal to the South China Sea. Another separated region of strong MJO signals is located in the eastern Pacific warm pool off the Central American coast. Similarly, Figure 3b shows that the unpredictable variability in JJA also has a secondary peak north of the equator from the Bay of Bengal to the South China Sea (with standard deviation 1.5 mm/day) and a second center in the eastern Pacific warm pool off the Central American coast (with standard deviation 1.0 mm/day). This is also the case for the standardized unpredictable covariance for the global precipitations with the MJO index (Figure 4b) for which JJA also has a secondary peak north of the equator from the Bay of Bengal to the South China Sea, but with much weaker strength (with standard deviation 0.8–1.0 mm/day). There is also a second center in the eastern Pacific warm pool off the Central American coast, but with weaker strength (with standard deviation 0.8 mm/day). Figure 3c shows the unpredictable variability in SON with a similar pattern to that of JJA, but with less variability (Figure 3c), and this is also the case for the standardized unpredictable covariance for the MJO index with tropical ocean precipitations. This coincides with the fact that September also belongs to the second peak season of MJO.

Zhang and Dong [8] suggested that for the broad tropical region, the seasonality in the MJO is featured by a latitudinal migration across the equator between two peak seasons. In particular, MJO signals in the eastern Pacific (Maloney and Kiehl [9]) appear to be much stronger during boreal summer than winter. Such migration can be clearly seen for its unpredictable variability. In JJA and SON (Figure 3b,c), the 1.0 mm/day contour of the unpredictable standard deviation reaches north of Japan close to 50° N, while in DJF it draws back to north of the Philippines about 15 N. Similarly, in DJF and MAM (Figure 3a,d), the 1.0 mm/day contours reach north of New Zealand close to 30° S, while in JJA it draws back to north of Australia about 10 S. However, these are not obvious in the standardized unpredictable covariance for the MJO index with global precipitation. The pattern correlations between the unpredictable component variability (Figure 3) and the standardized unpredictable covariance of the precipitations with the MJO index (Figure 4) in tropical oceans are 0.79, 0.79, 0.77 and 0.84 in MAM, JJA, SON and DJF, respectively. The corresponding explained unpredictable variances by MJO-PC1 and MJO-PC2 are around 0.065 for all the seasons.

### 3.3. Potential Predictability

The spatial patterns of the potential predictability in the four seasons are shown in Figure 5. In tropical oceans, they are more similar to the predictable variability (Figure 1) than the unpredictable variability (Figure 2). In particular, the potential predictability along the tropical Pacific is lowest in JJA (Figure 5b) associated with the lowest predictable variability (Figure 1b). Potential predictability is highest in DJF (Figure 5d) associated with the highest predictable variability (Figure 4d). Predictability along the tropical Atlantic is the highest in MAM and lowest in SON (Figure 1a,c), which is consistent with the potential predictability (Figure 5a,c).

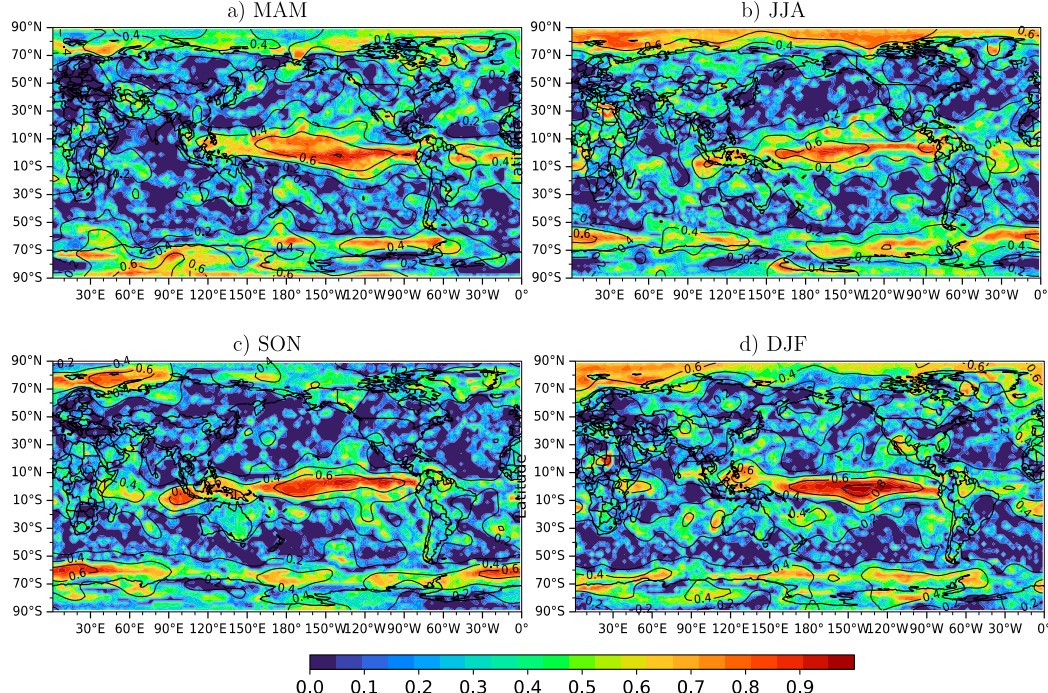

**Figure 5.** Seasonal potential predictability of GPCP precipitation.

The potential predictability of precipitations in the subtropics may also be related to ENSO. For example, the JJA and DJF precipitation of New Zealand has potential predictability around 0.3 (Figure 5b,d). Zheng and Frederiksen [10] further extended the decomposition methodology from the variability of single time series to the covariability of multivariate time series and established a statistical prediction scheme based on the prediction of EOFs of the covariance matrix of the predictable components. They showed that the predictability is closely related to Nino3 SST, but also related to the tropical Indian Ocean SST, the local NZ SST and the southern annular mode, with the cross-validated explained variance about 20%. For another example, the potential predictability of the eastern Asian summer monsoon rainfall [the seasonal mean rainfall in the region 5–50° N, 100–140° E in June-July-August which includes the Meiyu belt from Southern Japan through the Yangzi River catchment] is around 0.3 (Figure 5b). By using the methodology of Zheng and Frederiksen [10], Yin et al. [11] demonstrated that the predictability is not only related to the ENSO decay and developing phases, but also to seven other predictors from the Pacific, Atlantic and Indian Oceans, making the cross-validated explained variance more than 0.25, which reaches about 80% of the potential predictability.

Figure 5 also shows that there are some areas with significant potential predictability in some seasons, for example, the western Mediterranean centered in northern Egypt in JJA (Figure 5b) and northern Mexico in DJF (Figure 5d). To our knowledge, no study similar to Zheng and Frederiksen [10] has been carried out for precipitation in these regions. However, the potential predictability study in the current paper at least provides clues as

to where the precipitation is more likely to be predicted and suggests that the statistical prediction scheme proposed by Zheng and Frederiksen [10] is worth attempting.

## 4. Discussions on Methodology

In earlier studies, daily time series were used to estimate the unpredictable weather noise component. A critical step is to separate the potentially predictable variability from daily data. For continuous metrological variables, such as temperature and pressure, Madden [1] applied a frequency domain approach where the potentially predictable variability was removed by a Fourier transform on daily time series for all seasons. Shukla [12] and Trenberth [13] applied an alternative time domain approach by assuming day-to-day weather variability is red noise, and the potentially predictability component was partially removed by a non-parametric approach in estimating the variability of the weather noise component. The red noise constraint was further relaxed as colored noise, and the influence of the potentially predictable component was completely removed by the application of the first order difference operator (Zheng [14]) or modeled as a variance parameter (Jones [15]) or multi-year mean parameters (Delsole and Feng [16]). However, daily rainfall data are intermittent and therefore the methodologies for daily continuous metrological data are no longer suitable. Madden et al. [17] applied the chain-dependent model (Katz [18]) to estimate weather noise variability, but for the daily precipitation observations, the fitted parameters are dependent on the predictable components (e.g., Katz and Zheng [19]; Zheng et al. [20]) and therefore the model is over simplified. Feng and Houser [21] modeled the daily precipitation as Gaussian auto-regressive processes described by Delsole and Feng [16], but the daily precipitation may not be Gaussian, especially if there are considerable dry days in small areas.

Zheng et al. [7] developed a method for estimating the weather noise component only, using monthly mean time series based on the likelihood estimation assuming the monthly means of the weather noise components are Gaussian. They demonstrated, by simulation, that the estimations of the weather noise component are comparable to those using daily continuous meteorological data. Zheng and Frederiksen [3] extended the methodology of Zheng et al. [7] for estimating the covariability of the potentially predictable components based on the moment estimation with a relaxed Gaussian assumption. Since the Gaussian assumption on the monthly mean of the unpredictable component is no longer required, the methodology can, in principle, be applied to precipitation data.

In this paper, the methodology of Zheng and Frederiksen [3] was applied to analyze the potential predictability of the GPCP precipitation. There are a number of advantages of the methodology applied in this study over all the previous methodologies using daily data. Monthly data are more available than daily data. This methodology can be applied to estimate the covariability of the predictable (unpredictable) components between two climate variables, which allows for the influence of one climate variable on another to be estimated, as shown in this study. Moreover, it can be used to estimate the covariance matrix of predictable (unpredictable) components of a climate field to which the singular value decomposition analysis can be applied on the estimated component covariance matrices. Such an approach was successfully applied to the seasonal prediction of New Zealand rainfall (Zheng and Frederiksen [10]) and east Asia summer monsoon rainfall (Ying et al. [11]). The successful predictions indicate that the estimation method based on monthly rainfall data is reliable. Other existing methods can only be applied to estimate the potential predictability of a single climate variable, but not the covariability between two climate variables.

By using monthly data, fewer statistical assumptions are required compared with methodologies using daily data. The only assumption is that the variance of $\varepsilon_{ym}$ is invariant for month $m$, while the estimations using daily data depend on a variety of stationary assumptions and parameterizations which can lead to quite different results. For example, comparing the estimated potential predictabilities estimated in this paper (Figure 5) and those using the methodology of Delsole and Feng [16] for the GPCP data (see Figure 1 of

Feng and Houser [21]), although there is much higher predictability for the South Pole in their study, a higher predictability at the North Pole during JJA in our study is not shown in their study. Furthermore, the potential predictability of the eastern Asia summer monsoon rainfall is 30% using our method, while it is nearly zero in their study. Using our method, Ying et al. [11] developed a seasonal prediction scheme with a 0.25 cross-validated explained predictability for the eastern Asia summer monsoon rainfall.

## 5. Discussion and Conclusions

The covariance decomposition method proposed by Zheng and Frederiksen [3] was applied to the monthly GPCP precipitation data. The variances of the precipitation seasonal means for the four seasons were decomposed into potentially predictable components driven by slowly varying boundary forcing and low-frequency internal dynamics, and the unpredictable component related to day-to-day weather variability. It was suggested that ENSO (represented by Nino3-4 SST) contributes more than half of the predictable variability in SON and DJF when strong and unlikely to change phase. Although ENSO in JJA is weaker than ENSO in MAM, it contributes about 40% of the predictable variability in JJA compared with about 25% in MAM. This could be due to the fact that ENSO in MAM is more likely to change phase, especially in April. The seasonal distribution of the variability in the unpredictable component of GPCP precipitation is similar to that of the MJO described by Zhang and Dong [8], though less than 7% of the variability in the unpredictable component of GPCP precipitations was explained by the MJO index. This raises the question as to whether the current MJO indexes represent MJO variability well, or the MJO may not be so dominant in the variability of the unpredictable component of seasonal precipitation compared to ENSO and, therefore, this subject is worth further investigation.

**Author Contributions:** Conceptualization, H.L. and X.Z.; methodology, X.Z. and C.S.F.; formal analysis, H.L., X.Z. and J.Y.; data curation, J.Y.; writing—original draft preparation, X.Z and H.L.; writing—review and editing, C.S.F. All authors have read and agreed to the published version of the manuscript.

**Funding:** This research received no external funding.

**Data Availability Statement:** The data used in this study are available publicly. The GPCP precipitation is available at: https://psl.noaa.gov/data/gridded/data.gpcp.html (accessed on 6 April 2023). The MJO index is available at: https://www.psl.noaa.gov/mjo/mjoindex/ (accessed on 6 April 2023).

**Acknowledgments:** We sincerely thank the reviewers for their comments to improve the paper.

**Conflicts of Interest:** The authors declare no conflict of interest.

## Appendix A

R-code for calculating the unpredictable covariance of Equation (4)

```
Cov.wmr=function(x1, x2) #x1,x2: Y by 3 matrix of the two monthly data sets
{
m <- dim(x1) [1]
X1 <- x1[, 1:2] − x1[, 2:3]
X2 <- x2[, 1:2] − x2[, 2:3]
a <- (sum(X1[, 1] * X2[, 1]) + sum(X1[, 2] * X2[, 2]))/2/m
b <- (sum(X1[, 1] * X2[, 2]) + sum(X1[, 2] * X2[, 1]))/2/m
alpha <- min(0.1, max((a + 2 * b)/(a + b)/2, 0))
sigma <- a/(1-alpha)/2
(sigma * (3 + 4 * alpha))/9
}
```

**Appendix B**

Proof of the variance of $\mu_y$ ($\varepsilon_{yo}$) explained by $\mu_y'$ ($\varepsilon_{yo}'$).

We only prove for predictable component; proof for unpredictable component is similar.

Suppose the numerical values of $\mu_y$ and $\mu_y'$ exist, and $\mu_y$ can be regressed by $\mu_y'$ as

$$\mu_y = b\mu_y' + \xi_y$$

where $b$ is the regression coefficient and $\xi_y$ is the regression error. Since $\xi_y$ and $\mu_y'$ are statistically independent, then

$$V\left(\mu_y, \mu_y'\right) = bV\left(\mu_y', \mu_y'\right)$$

so the regression coefficient $b$ can be estimated as

$$b = V\left(\mu_y, \mu_y'\right) / V\left(\mu_y', \mu_y'\right)$$

therefore, the variance of $\mu_y$ explained by $\mu_y'$ is

$$V\left(b\mu_y', b\mu_y'\right) = b^2 V\left(\mu_y', \mu_y'\right) = V^2\left(\mu_y, \mu_y'\right) / V\left(\mu_y', \mu_y'\right)$$

Although the numerical values of $\mu_y$ and $\mu_y'$ cannot be obtained, $V(\mu_y, \mu_y')$ can be estimated using Equation (3).

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
