# Peer review of "Potential Predictability of Seasonal Global Precipitation Associated with ENSO and MJO"

_atmosphere, doi:10.3390/atmos14040695_

Round 1
Reviewer 1 Report
Overall, I found this paper to be interesting. The methods are well supported by the literature. The results are clear, and the conclusions follow the results. The presentation of the work is very good, and I just have minor edits and suggestions for the authors.
Line 30: seasonal doesn’t need to be capitalized here.
Line 163: can you tell us what this standard deviation is measured in? It wasn’t clear if the scale was in mm per day similar to Figure 3, and I only knew that because of the mention of this on line 215. I would be sure to specify this in the captions for Figures 1 and 3.
Line 172: I would capitalize the F in Figure here.
Line 189: Could you add an introduction to this section similar to what you did for ENSO on lines 150-153. The introduction of that section showed right away that the potential predictability was high (more than 0.9) which then you built the rest of the section on. Here, you are saying that the potential predictability is low for MJO, but can you tell us how low, or how that compares to ENSO?
Line 212: I would capitalize the F in Figure here as well. I also note that sometimes the word “figure” is in bold and other times it is not, so I would change that so it is consistent throughout.
Line 290-291: The text looks smaller here, either the font changed or the spacing? Also, I think it would sound a bit better to say “is worthwhile to try” rather than “is worth to try”.
Line 301: what is meant by red here?
Lines 412-413: There is some text that is out of place here that seems to be talking about what is appropriate for the appendix. Maybe leftover from a template?
Author Response
Dear Reviewer one,
Thank you very much for commenting on our manuscript. Indeed it leads to an improved paper. Thanks again.
Here is the point-to-point response.
Overall, I found this paper to be interesting. The methods are well supported by the literature. The results are clear, and the conclusions follow the results. The presentation of the work is very good, and I just have minor edits and suggestions for the authors.
Line 30: seasonal doesn’t need to be capitalized here.
Thank you for pointing out this. Changed to lower case.
Line 163: can you tell us what this standard deviation is measured in? It wasn’t clear if the scale was in mm per day similar to Figure 3, and I only knew that because of the mention of this on line 215. I would be sure to specify this in the captions for Figures 1 and 3.
Yes, added the units mm/day for Figure 1 and 3 captions.
Line 172: I would capitalize the F in Figure here.
Sure, now it is (Figure 1)
Line 189: Could you add an introduction to this section similar to what you did for ENSO on lines 150-153. The introduction of that section showed right away that the potential predictability was high (more than 0.9) which then you built the rest of the section on. Here, you are saying that the potential predictability is low for MJO, but can you tell us how low, or how that compares to ENSO?
Thanks for the comment. It is revised as “The Madden-Julian Oscillation (MJO) is a tropical weather system originating from Indian Ocean that propagates eastward around the global tropics with a cycle on the order of 30-90 days [7]. The MJO indices MJO-PC1 and MJO-PC2 are the projections of 20-96 day filtered OLR [8], the seasonal potential predictability of the MJO-PC1 and MJO-PC2 are virtual zero. Thus, the MJO is more likely to be a driver of the unpredictable variability of the tropical precipitation. Therefore, any global precipitation related to the two MJO indices is also likely to be highly unpredictable.”
Line 212: I would capitalize the F in Figure here as well. I also note that sometimes the word “figure” is in bold and other times it is not, so I would change that so it is consistent throughout.
Yes, changed all to bold and capitalized the F.
Line 290-291: The text looks smaller here, either the font changed or the spacing? Also, I think it would sound a bit better to say “is worthwhile to try” rather than “is worth to try”.
Yes, changed accordingly to is worthwhile to try.
Line 301: what is meant by red here?
Yes, it was not clear. Changed to red noise.
Lines 412-413: There is some text that is out of place here that seems to be talking about what is appropriate for the appendix. Maybe leftover from a template?
Yes, deleted those left over from the template. Sorry for leaving it out there.
Reviewer 2 Report
- To explain the seasonal patterns of time series, it is necessary and necessary to explain, measure, analyze, predict, and control the processes that unfortunately have not been specified in the compilation of this article, so the conclusion in this direction will be incorrect.
- It is very important to decompose and separate temporal and spatial patterns, when seasonal patterns are discussed, this differentiation of series should be done in terms of time and space, which was not done in this article.
- In the explanation of the topic, the spatial explanation is based on the explanation of the time pattern of the series, which causes problems in the methodology of the research stages. In line with this methodology, the results will be misinterpreted.
- When it comes to the explanation of seasonal patterns, the use of seasonal cycles by the temporal explanation of the series will lead to errors in the separation of temporal patterns, and this pattern of explanation can be seen in the research in the article.
- In the article, there is no validation method or model in explaining seasonal stages and separating time patterns.
- The references used in the article are not up-to-date and have a suitable variety.
- The inappropriate explanation of the periods of rainfall series is one of the other points of this article and the reason for using daily, monthly, and seasonal series is not used.
Author Response
Dear Reviewer,
Thank you for commenting on our manuscript. We appreciate your thoughts on time series data analysis.
The potential predictability indicates how better a variable can be predicted skillfully. So the predictability research can help guide the forecast research. The weather/climate system is chaotic, which is influenced by many factors and hard to predict. Here we decompose the seasonal precipitation time serious into potentially predictable and unpredictable parts. Further to reduce the dimension of the predictable part we used the SVD decomposition technique. We calculated the covariance of the precipitation predictable part and ENSO index to tie the seasonal precipitation predictability with the ENSO. MJO index is widely used in the sub-seasonal forecast of precipitation, but it cannot be used for seasonal forecast. However, our method finds that the MJO indices can only play a limited role in unpredictable variability of seasonal precipitation. This rise the question whether the current MJO indexes well represent MJO variability, or the MJO may not be so dominant in the variability of the unpredictable component of the seasonal precipitation.
Round 2
Reviewer 2 Report
Validation of data and models has not been explained in detail yet
Author Response
Dear Reviewer,
Thank you for the comment asking for more details on the data and the model. The revised Data and Model part is attached in a separate file.
Thank you again.
